# Childhood Maltreatment in Females Is Associated with Enhanced Fear Acquisition and an Overgeneralization of Fear

**DOI:** 10.3390/brainsci12111536

**Published:** 2022-11-12

**Authors:** Phillip Zoladz, Kassidy Reneau, Jordan Weiser, Chloe Cordes, Emma Virden, Sara Helwig, Caitlin Thebeault, Cassidy Pfister, Bruktawit Getnet, Kayla Boaz, Taylor Niese, Mercedes Stanek, Kristen Long, Sydney Parker, Boyd Rorabaugh, Seth Norrholm

**Affiliations:** 1Psychology Program, The School of Health and Behavioral Sciences, Ohio Northern University, Ada, OH 45810, USA; 2Department of Pharmaceutical Sciences, Marshall University School of Pharmacy, Huntington, WV 25755, USA; 3Department of Psychiatry and Behavioral Neurosciences, Wayne State University School of Medicine, Detroit, MI 48202, USA

**Keywords:** childhood maltreatment, stress, fear conditioning, generalization, startle response, sex differences

## Abstract

Childhood maltreatment may alter fear neurocircuitry, which results in pathological anxiety and depression. One alteration of fear-related behaviors that has been observed in several psychiatric populations is an overgeneralization of fear. Thus, we examined the association between childhood maltreatment and fear generalization in a non-clinical sample of young adults. Two hundred and ninety-one participants underwent differential fear conditioning in a fear-potentiated startle paradigm. One visual stimulus (CS+), but not another (CS−), was associated with an aversive airblast to the throat (US) during acquisition. The next day, participants were tested for their fear responses to the CS+, CS−, and several generalization stimuli (GS) without the presence of the US. Participants also completed questionnaires that assessed symptoms of childhood maltreatment, anxiety, depression, and post-traumatic stress disorder (PTSD). Participants reporting high childhood maltreatment (*n* = 71; 23 males, 48 females) exhibited significantly greater anxiety, depression, and symptoms of PTSD than participants reporting low childhood maltreatment (*n* = 220; 133 males, 87 females). Females reporting high childhood maltreatment demonstrated significantly enhanced fear learning and greater fear generalization, based on their fear-potentiated startle responses. Our findings suggest that childhood maltreatment may sex-dependently influence the development of fear neurocircuitry and result in greater fear generalization in maltreated females.

## 1. Introduction

Nearly 10% of children in Western societies endure some form of maltreatment during any given year [1]. Childhood maltreatment, which involves verbal, physical, or sexual abuse or significant physical or emotional neglect in youth, is one of the strongest risk factors for, and the leading preventable cause of, many psychological disorders [1,2,3,4,5,6]. It is also associated with increased suicide attempts [7] and non-suicidal self-injury [8], as well as a greater risk for the development of several physiological ailments, including cardiovascular disease, diabetes, and stroke [9].

Researchers have speculated that childhood maltreatment increases the susceptibility for psychological disorders by exerting deleterious effects on the development of brain regions involved in emotion, particularly fear [10,11,12,13]. Developmental alterations of these brain areas may promote stronger responses to threatening stimuli, as well as abnormal fear learning, which may ultimately foster pathological anxiety and depression. Consistent with this speculation, adults with a history of childhood maltreatment exhibit reduced size of limbic brain regions, including the amygdala, hippocampus, and prefrontal cortex (PFC) [14,15,16,17,18,19,20,21,22,23,24,25,26], as well as decreased frontolimbic functional connectivity [27,28,29,30]. The PFC has a well-established role of inhibiting inappropriate emotional responses and is important for the extinction of conditioned fear [31]. The hippocampus is vital for contextual fear learning and has been implicated in stimulus discrimination and distinguishing between the threat/safety of stimuli [32,33]. Smaller volumes, and conceivably impaired functioning, of these structures could therefore result in hyperactive emotional brain areas (e.g., amygdala) and lead to enhanced fear learning [34]. Indeed, research has shown that childhood maltreatment is associated with quicker responses, greater attention, and heightened amygdala reactivity to emotional stimuli [30,35,36,37,38,39].

Although childhood maltreatment results in greater behavioral and neural responses to emotional stimuli, little research has studied its association with fear learning. In one of the few studies that has examined this relationship, McLaughlin and colleagues found, somewhat counterintuitively, that maltreated children aged 6–18 years exhibited blunted skin conductance responses (SCRs) to a CS+ and smaller amygdala and hippocampal volumes, relative to non-maltreated children [11]. Still, amygdala and hippocampal volume correlated positively with SCRs to the CS+ in maltreated children. According to the investigators, the findings suggested that maltreated children exhibit impaired associative learning or overgeneralize their fear of the CS+ to other stimuli.

The available empirical literature suggests that overgeneralizing fear is a robust pathogenic marker of psychopathology [40,41,42,43]. Stimulus generalization involves an organism exhibiting a conditioned response (CR) to stimuli that are perceptually or semantically similar to a conditioned stimulus (CS). Generalization of fear is adaptive because it allows an organism to generalize fear to novel stimuli that may present a threat to survival. However, broad generalization can be burdensome to daily life and promote fear in response to non-threatening stimuli. A few studies have examined the impact of childhood maltreatment on fear generalization, but the results of these studies have been inconclusive and accompanied by caveats. Lange and colleagues found that maltreated youth aged 16–25 displayed greater generalization following differential fear conditioning than non-maltreated youth, as measured by subjective fear ratings and US expectancy ratings [12]. This effect was particularly evident in maltreated youth who exhibited a greater number of subclinical symptoms of anxiety, depression, and psychosis and was associated with reduced hippocampal activity during the task. In addition, Thome and colleagues [44] used fear-potentiated startle to assess fear generalization in a sample of adult females diagnosed with post-traumatic stress disorder (PTSD) as a result of repeated childhood abuse. The results of this study provided some, albeit minimal, evidence for greater generalization in PTSD patients. Specifically, these individuals exhibited a smaller increase in reaction time in response to generalization stimuli than did control subjects, suggesting greater uncertainty about the threat/safety of the stimuli. There was, however, no significant impact of PTSD on generalization as measured by fear-potentiated startle.

Given the limited work in this area, the purpose of the present study was to assess the impact of childhood maltreatment on differential fear conditioning and fear generalization in a non-clinical sample of young adults using established psychophysiological methods (e.g., [45,46]). Previous work has demonstrated that childhood maltreatment results in physiological and behavioral changes that lead to perturbed fear circuitry. Thus, abnormal acquisition and/or expression of fear may mediate or moderate the relationship between childhood maltreatment and psychopathology. We therefore hypothesized that young adults reporting childhood maltreatment would exhibit stronger fear learning and greater fear generalization.

## 2. Materials and Methods

### 2.1. Participants

The data presented in this paper are a subset of data from a larger study examining the influences of stress, sex, and childhood maltreatment on fear generalization. Two hundred and ninety-one healthy undergraduate students (156 males, 135 naturally cycling females; age: *M* = 19.37, *SD* = 1.93) from Ohio Northern University volunteered to participate in the experiment. Individuals were excluded from participating if they met any of the following conditions: diagnosis of post-traumatic stress disorder (PTSD); presence of skin diseases, such as severe psoriasis, eczema, or scleroderma; history of syncope or vasovagal response to stress; history of any heart condition or cardiovascular issues (e.g., high blood pressure); history of severe head trauma; current treatment with narcotics, beta-blockers, or steroids; pregnancy; substance use disorder; regular use of recreational drugs; regular nightshift work; hearing loss. Participants were asked to refrain from drinking alcohol or exercising extensively for 24 h prior to the experimental sessions and to refrain from eating or drinking anything but water for 2 h prior to the experimental sessions. These exclusion criteria were utilized to prevent adverse effects of the fear learning procedure on the physical and/or psychological health of participants; they were also employed to mitigate the impact of confounding variables (e.g., alterations of circulating hormone levels due to ingestion of food products) on the dependent measures.

### 2.2. Ethical Considerations

All experimental procedures were approved by the Institutional Review Board at Ohio Northern University, carried out in accordance with the Declaration of Helsinki, and undertaken with the understanding and written consent of each participant. Participants were compensated with class credit and $20 USD upon completion of the study.

### 2.3. Differential Fear Conditioning Paradigm

On Day 1, participants underwent an acquisition phase and were trained to associate one conditioned stimulus (CS+) with an aversive unconditioned stimulus (US), while also learning that a different conditioned stimulus (CS−) was not associated with the US. On Day 2, participants underwent the generalization phase of the paradigm and were exposed to the CS+, CS−, and several generalization stimuli (GS) without any presentation of the US, in a manner adapted from previous work [47,48]. During each phase, fear-potentiated startle was the primary dependent measure.

#### 2.3.1. Conditioned, Unconditioned, and Generalization Stimuli

The CSs and GSs consisted of nine black circles of increasing diameter (see Figure 1), which were presented on a white background of a computer monitor (via SuperLab software; Cedrus Corporation, San Pedro, CA, USA). The monitor was located approximately 124 cm in front of participants. The smallest circle was 8.5 mm in diameter, and each larger circle increased in size by 2.5 mm, ending with the largest circle, which was 29 mm in diameter. For approximately half of the participants (*n* = 148), the smallest circle served as the CS+, and the largest circle served as the CS−. For the remaining participants (*n* = 143), the largest circle served as the CS+, and the smallest circle served as the CS−. The circles sized between the CS+ and the CS− were the generalization stimuli (GS) and were only presented to participants on Day 2.

The unconditioned stimulus (US) was a 250-ms, 140-p.s.i. airblast aimed at the larynx. A similar airblast has been used in previous work and reliably generates fear-potentiated startle (e.g., [45,46,49,50,51,52,53,54,55]). The startle probe was a 40-ms, 108-dB burst of white noise that was delivered to participants through a pair of headphones. During CS+ trials on Day 1, the startle probe was presented 6 s after the onset of the CS, which was followed 500 ms later by presentation of the US; the CS+ terminated 500 ms after the onset of the US. During all GS and CS− trials (and CS+ trials on Day 2), the startle probe was presented 6 s following the onset of the stimulus, without any presentation of the US; stimulus presentation ended 250 ms after the startle probe. On noise alone (NA) trials, only the startle probe was presented while participants looked at the white background on the computer monitor; the length of NA trials matched the length of the startle probe (i.e., 40 ms).

#### 2.3.2. Startle Response Measurement

Electromyographic (EMG) recordings of the right orbicularis oculi muscle were used to measure participants’ startle responses. Ag/AgCl electrodes (5-mm) that were filled with electrolyte gel were placed 1 cm below the pupil of the right eye, 1 cm below the lateral canthus, and over the mastoid behind the right ear (ground). The impedance levels for each participant were less than 6 kΩ. The EMG data were obtained via the EMG module of the Biopac MP150 system (Biopac Systems, Inc., Aero Camino, CA, USA) and the Acqknowledge data acquisition and analysis software (Biopac Systems, Inc., Aero Camino, CA, USA). The EMG signal was sampled at 1 kHz.

#### 2.3.3. US Expectancy Measurement

Participants used a three-button response keypad (SuperLab software, Cedrus Corporation, San Pedro, CA, USA) to report their expectation of the US during each CS+, GS, and CS− trial on Days 1 and 2. When each stimulus was presented on the computer monitor, participants pressed an “AIR” key if they expected the US to occur, a “NO AIR” key if they did not expect the US to occur, and a “?” key if they were unsure if the US would occur. For data analysis, “AIR” responses were scored as +1, “?” responses were scored as 0, and “NO AIR” responses were scored as −1 [45,46,49,50,51,52,53,54,55].

#### 2.3.4. Fear Acquisition (Day 1) and Fear Generalization (Day 2)

On Day 1, participants completed the acquisition phase of the paradigm. Acquisition began with three NA trials, followed by a habituation phase that included four NA, CS+, and CS− trials. The CS presentations during habituation were not followed by the US. After habituation, participants underwent the conditioning phase, which included three blocks of four trials of each stimulus type (NA, CS+, CS−); this resulted in 12 trials per block and 36 total trials. During conditioning, the CS+ was always paired with the US (i.e., 100% reinforcement rate), as in prior studies with this paradigm (for example, see [56]). 

On Day 2, participants completed the generalization phase of the paradigm. This phase began with three NA trials, followed by three blocks of one trial of each stimulus type (NA, CS+, seven GSs, CS−) for a total of 10 trials per block and 30 total trials. No CS or GS presentation during the generalization phase was reinforced with the US.

Fixed trial orders were used for all participants. The only restrictions were that, during acquisition, there were 4 trials of each trial type (i.e., NA, CS+, CS−) for each block of 12 trials and, during generalization, there was 1 trial of each trial type (i.e., NA, CS+, seven GSs, CS−) for each block of 10 trials. The fixed trial order involved randomizing the trial types within each block. The intertrial interval was random, between 9 and 22 s in duration. Figure 1 depicts the timeline, stimuli, and trial block composition that made up each experimental session.

#### 2.3.5. Startle Data Preprocessing

The Acqknowledge data files were imported into the MindWare EMG analysis program (MindWare Technologies, Ltd., Gahanna, OH, USA). This program was used to filter and rectify the EMG signals that occurred 20–200 ms following the presentation of each startle probe. The EMG signal was amplified with a gain of 2000. It was filtered with low- and high-frequency cutoffs at 28 and 500 Hz, respectively, and a 60-Hz notch filter was applied. The resulting data were exported for analysis. The peak EMG signal 20–200 ms after each startle probe was used as a measure of the startle response.

### 2.4. Questionnaires

Following the generalization phase of the fear conditioning paradigm on Day 2 (i.e., upon completing all experimental manipulations involved in the study), participants completed several questionnaires to assess symptoms of childhood maltreatment, anxiety, PTSD, and depression. The questionnaires related to anxiety, PTSD, and depression were included to verify that participants who reported high childhood maltreatment would also exhibit greater symptoms of current psychological distress.

#### 2.4.1. Childhood Trauma Questionnaire (CTQ)

The CTQ is a 28-item questionnaire that retroactively quantifies childhood maltreatment and results in 5 subscale scores, ranging from 5–25, for emotional abuse, physical abuse, sexual abuse, emotional neglect, and physical neglect. Bernstein and Fink [57] established scores for none, mild, moderate, and severe levels of each type of childhood abuse and neglect. Similar to previous work [58,59,60], we divided participants into “high childhood maltreatment” and “low childhood maltreatment.” Participants experiencing moderate or severe levels in *any category* of abuse or neglect (emotional abuse >12; physical abuse >9; sexual abuse >7; emotional neglect >14; physical neglect >9) were assigned to the high childhood maltreatment category (*n* = 71; 23 males, 48 females); participants experiencing none or mild levels in *all categories* of abuse and neglect (emotional abuse <13; physical abuse <10; sexual abuse <8; emotional neglect <15; physical neglect <10) were assigned to the low childhood maltreatment category (*n* = 220; 133 males, 87 females). In the current sample, the CTQ data demonstrated excellent reliability (Cronbach’s alpha = 0.82).

#### 2.4.2. State-Trait Anxiety Inventory (STAI)

The STAI is a 40-item questionnaire that assesses symptoms of anxiety by having participants indicate how they feel at the time they are completing the questionnaire (i.e., “state” anxiety) and how they “generally feel” (i.e., “trait” anxiety) [61]. Each subscale ranges from 20–80. In the current sample, the STAI data demonstrated excellent reliability (Cronbach’s alpha = 0.94).

#### 2.4.3. Anxiety Sensitivity Index—Version 3 (ASI-3)

The ASI-3 is an 18-item questionnaire that assesses symptoms of anxiety sensitivity by quantifying participants’ tendencies to misinterpret anxiety-related sensations [62]. The score ranges from 0–72. In the current sample, the ASI-3 data demonstrated excellent reliability (Cronbach’s alpha = 0.88).

#### 2.4.4. PTSD Checklist—Civilian Version (PCL-C)

The PCL-C is a 17-item questionnaire that assesses symptoms of PTSD in participants. In particular, the questionnaire measures symptoms of avoidance, hyperarousal, anhedonia, anxiety, intrusive memories, and sleep disturbances [63]. It produces a score ranging from 17–85. In the current sample, the PCL-C data demonstrated excellent reliability (Cronbach’s alpha = 0.90).

#### 2.4.5. Center for Epidemiologic Studies Depression Scale (CES-D)

The CES-D is a 20-item questionnaire that assesses symptoms of depression in participants (e.g., depressed mood, anhedonia, helplessness) and ranges from 0–60 [64]. In the current sample, the CES-D data demonstrated excellent reliability (Cronbach’s alpha = 0.90).

### 2.5. Statistical Analyses

Two-way ANOVAs were used to analyze the state and trait components of the STAI, the ASI-3, the PCL-C, and the CES-D, with childhood maltreatment (high, low) and sex (male, female) as the between-subjects factors. Similar to previous work employing the fear-potentiated startle paradigm (e.g., [46,47,50,51,52,54]), we quantified fear-potentiated startle by computing a difference score for the EMG recordings [(startle magnitude to the CS+, CS−, or GS in each block)—(startle magnitude to the NA trials in each block)]. Raw difference scores allowed us to calculate fear-potentiated startle responses based on each participant’s startle responses to the NA trials and is supported by research demonstrating their superiority to standardized difference scores and percent change scores [65]. Because of startle responses can vary largely from trial to trial, we calculated difference scores for each trial type within each block of acquisition on Day 1 (i.e., average of 4 trials of each trial type) to obtain a more accurate representation of fear-potentiated startle responses within each block [66,67,68].

Separate mixed-model ANOVAs were used to analyze baseline startle responses (i.e., responses to the first 3 NA trials), fear-potentiated startle responses, and US expectancies on Days 1 and 2, with childhood maltreatment and sex as the between-subjects factors. For the analyses of fear-potentiated startle and US expectancies during the acquisition phase on Day 1, stimulus (CS+, CS−) and trial block (4 blocks) served as the within-subjects factors. Late acquisition was defined as block 4 on Day 1, when the CS+-CS− discrimination was at maximum. For the analyses of fear-potentiated startle responses and US expectancies during the generalization phase on Day 2, stimulus (CS+, CS−, seven GSs) and trial (3 trials) served as the within-subjects factors.

Alpha was set at 0.05 for all analyses. When the omnibus *F* indicated the presence of a significant effect, we used Bonferroni-corrected post hoc tests to determine what groups differed significantly. If the assumption of sphericity was violated, we used Greenhouse-Geisser corrections to reduce the degrees of freedom accordingly.

## 3. Results

### 3.1. Self-Report Measures of Mood and Anxiety

Participants reporting high childhood maltreatment exhibited significantly greater state anxiety, *F*(1,287) = 7.74, *p* = 0.006, trait anxiety, *F*(1,287) = 8.18, *p* = 0.005, PTSD-related symptoms *F*(1,287) = 25.46, *p* < 0.001, and symptoms of depression, *F*(1,287) = 19.44, *p* < 0.001, than participants reporting low childhood maltreatment (see Table 1). Participants reporting high childhood maltreatment also exhibited greater anxiety sensitivity than participants reporting low childhood maltreatment, but this effect did not reach statistical significance, *F*(1,287) = 3.24, *p* = 0.073.

### 3.2. Fear-Potentiated Startle

#### 3.2.1. Day 1—Acquisition Phase

The analysis of baseline startle responses to the first three NA trials during acquisition revealed that females reporting high childhood maltreatment exhibited significantly greater startle responses than all other groups (main effect of childhood maltreatment: *F*(1,261) = 3.76, *p* = 0.054; main effect of sex: *F*(1,261) = 8.86, *p =* 0.003; Childhood Maltreatment × Sex interaction: *F*(1,261) = 5.78, *p* = 0.017; see Figure 2). To control for the impact of this difference on fear-potentiated startle responses, we included baseline startle responses as a covariate in the analysis of fear acquisition. This analysis revealed that, during the non-reinforced habituation segment (CS HAB) and the first block of reinforced acquisition trials (ACQ1), participants exhibited significantly greater fear-potentiated startle responses to the CS− than to the CS+. In the final two blocks of reinforced acquisition trials (ACQ2 and ACQ3), participants exhibited significantly greater fear-potentiated startle responses to the CS+ than to the CS−, demonstrating successful differential fear conditioning (main effect of trial block: *F*(3,777) = 2.63, *p* = 0.049; Stimulus × Trial Block interaction: *F*(2.90,751.91) = 8.11, *p* < 0.001). By late acquisition, females reporting high childhood maltreatment displayed significantly greater fear-potentiated startle responses to the CS+ than did females reporting low childhood maltreatment and males reporting high or low childhood maltreatment (Sex × Stimulus × Trial Block interaction: *F*(2.90,751.91) = 3.51, *p* = 0.016; Childhood Maltreatment × Sex × Stimulus × Trial Block interaction: *F*(2.90,751.91) = 2.95, *p* = 0.034).

#### 3.2.2. Day 2—Generalization Phase

Baseline startle responses to the first three NA trials on Day 2 revealed that females exhibited significantly greater startle responses to the first and third NA trials than did males (main effect of sex: *F*(1,268) = 6.73, *p =* 0.01; main effect of trial: *F*(2,536) = 6.89, *p =* 0.001; Sex × Trial interaction: *F*(2,536) = 4.99, *p* = 0.007). Overall, participants reporting high childhood maltreatment tended to display greater baseline startle responses than participants reporting low childhood maltreatment, but this effect did not reach statistical significance (main effect of childhood maltreatment: *F*(1,268) = 3.78, *p =* 0.053). 

To control for the impact of baseline startle differences on fear-potentiated startle responses, we included baseline startle responses as a covariate in the analysis of fear generalization. This analysis of fear generalization revealed that participants exhibited significantly greater fear-potentiated startle responses to the CS+ than to any other stimulus (main effect of stimulus: *F*(7.30,3677.82) = 3.42, *p* < 0.001; see Figure 3). Similarly, participants displayed significantly greater fear-potentiated startle responses to GS1 than to GS2, GS4, GS5, GS7, and the CS−. These effects became more apparent during the second and third trial blocks (main effect of trial: *F*(1.94,519.08) = 4.31, *p* = 0.014; Stimulus × Trial Block interaction: *F*(13.78,3677.82) = 2.07, *p* = 0.007). Most importantly, the non-linear cubic component of the Childhood Maltreatment × Sex × Stimulus interaction was significant, *F*(1,267) = 3.88, *p* = 0.05. Thus, we performed Bonferroni-corrected post hoc tests to assess the simple effects of this interaction. Females reporting high childhood maltreatment demonstrated significantly greater fear-potentiated startle responses to the CS− than did females reporting low childhood maltreatment. There was also evidence of wider generalization gradient in females reporting high childhood maltreatment. For instance, these participants displayed statistically equivalent fear-potentiated startle responses to the CS+, GS1, GS2, GS3, GS5, GS6, and CS−. In sharp contrast, females reporting low childhood maltreatment (CS+ > GS2, GS4, GS5, GS7, CS−) and males reporting high (CS+ > GS1, GS2, GS4, GS5, GS6, GS7, CS−) or low (CS+ > GS2, GS3, GS4, GS5, GS6, GS7, CS−) childhood maltreatment exhibited significantly greater fear-potentiated startle responses to the CS+ than to several other stimuli, including the CS−.

### 3.3. US Expectancies

#### 3.3.1. Day 1—Acquisition Phase

During the non-reinforced habituation segment (CS HAB) of acquisition, US expectancy ratings for the CS+ were statistically equivalent to the US expectancy ratings for the CS−. However, during the first, second, and third blocks of reinforced acquisition trials, US expectancy ratings for the CS+ were significantly greater than US expectancy ratings for the CS−, and this difference increased as trial blocks progressed (main effect of stimulus: *F*(1,260) = 624.97, *p* < 0.001; main effect of trial block: *F*(1.62,422.13) = 10.24, *p* < 0.001; Stimulus x Trial Block interaction: *F*(2.22,576.60) = 278.45, *p* < 0.001; see Figure 4). Females reporting high childhood maltreatment also displayed greater US expectancy ratings for the CS+ and CS−, overall, than males reporting high childhood maltreatment (Childhood Maltreatment × Sex interaction: *F*(1,260) = 4.86, *p* = 0.028).

#### 3.3.2. Day 2—Generalization Phase

During generalization, US expectancy ratings for the CS+ and GS1 were significantly greater than US expectancy ratings for all other stimuli (main effect of stimulus: *F*(2.80,699.85) = 58.40, *p* < 0.001). These effects were most pronounced during the first trial block of generalization testing (main effect of trial block: *F*(1.64,410.63) = 21.54, *p* < 0.001; Stimulus × Trial Block interaction: *F*(9.74,2434.43) = 8.79, *p* < 0.001). There was a relatively smooth generalization gradient for the US expectancy ratings, and such ratings decreased as the stimuli transitioned from the CS+ toward the CS−. Females reporting high childhood maltreatment tended to report greater US expectancy ratings for the CS+ and GS1 than females reporting low childhood maltreatment, but this effect did not reach statistical significance.

## 4. Discussion

Childhood maltreatment is a significant predictor of psychological illness. Researchers have speculated that childhood maltreatment alters the development of fear neurocircuitry, which underlies the subsequent onset of pathological anxiety and depression. By altering the development of fear neurocircuitry, childhood maltreatment may lead individuals to exhibit stronger fear learning and to be fearful of stimuli that are not threatening, perhaps as a result of an overgeneralization of fear. In the present study, we examined the impact of childhood maltreatment on differential fear conditioning and fear generalization in a non-clinical sample by using an established fear-potentiated startle paradigm. Participants reporting high childhood maltreatment exhibited greater levels of anxiety, depression, and PTSD symptoms than participants reporting low childhood maltreatment. Most importantly, females, but not males, reporting high childhood maltreatment demonstrated stronger fear learning and greater fear generalization than participants reporting low childhood maltreatment. Such effects were evident for physiological (i.e., EMG-based fear-potentiated startle), but not subjective (i.e., US expectancy ratings), measures of fear. These results support the notion that childhood maltreatment alters the acquisition and expression of conditioned fear. They also suggest that fear neurocircuitry may be more susceptible to maltreatment-induced alterations in females.

We found that high childhood maltreatment was associated with greater baseline startle responses in females only. This is related to previous work from our laboratory [46] in which females exhibited significantly greater baseline startle responses than males (childhood maltreatment was not assessed in this previous work, however). In the present study, high childhood maltreatment was also associated with greater fear-potentiated startle responses to the CS+ during late acquisition in females, but not males, a finding consistent with the observation of enhanced fear-potentiated startle in females who have experienced psychological trauma [53]. Our findings appear to be inconsistent with those from McLaughlin and colleagues [11], who found that maltreated youth exhibited *blunted* SCRs to a CS+ in a differential fear conditioning paradigm. Methodological differences between our study and that of McLaughlin and colleagues could account for some of the discrepant results. While McLaughlin and colleagues studied youth between the ages of 6 and 18, we examined young adults 18 years of age and older. Moreover, McLaughlin and colleagues measured SCRs, whereas we utilized fear-potentiated startle (via EMG) as our primary measure of interest. Using fear-potentiated startle as a measure of fear provides a distinct advantage over the use of SCRs because, unlike SCRs, fear-potentiated startle is directly coupled with amygdala activity, provides a direct measure of fear, and is sensitive to stimulus valence [69,70].

To our knowledge, this is the first observation of a sex-dependent association of childhood maltreatment with fear learning in humans. However, preclinical work examining the impact of chronic stress on fear learning has also revealed sex-dependent effects. Specifically, chronic stress enhances fear acquisition in male rats [71,72,73], but can enhance [74] or impair [75] fear acquisition in females. Preclinical studies assessing the effects of early life stress on fear learning in adulthood are also suggestive of sex-dependent effects, with studies showing that early life stress enhances fear conditioning in adult males [76], while impairing it in adult females [77]. The sex-dependent effects of chronic stress on fear learning are likely attributable, at least in part, to the impact of sex hormones on emotional brain regions [31].

Consistent with our sex-dependent findings, some clinical research suggests that females are more sensitive to the impact of childhood maltreatment on neural connectivity. Herringa and colleagues [78] examined the relationship between childhood maltreatment and resting-state functional connectivity of the amygdala and hippocampus. Their results showed that maltreatment was associated with lower hippocampus-subgenual anterior cingulate cortex (sgACC) connectivity in both males and females, but it was associated with lower amygdala-sgACC connectivity in females only. The sgACC is a part of the ventromedial PFC, which aids in fear extinction by suppressing amygdala neural activity [31]. Uncoupling of sgACC-amygdala connectivity could result in impaired emotion regulation and an inability to adequately suppress fear responses. Because females, but not males, exhibited evidence for such uncoupling, maltreatment may result in more unregulated fear responses in females and lead to stronger fear learning.

The differential impact of childhood maltreatment on males and females may also relate to how they are socialized to express their emotions as children. For instance, extensive work suggests that girls are expected to display greater levels of emotions than boys [79,80], which could result in more overt fear-related behaviors in girls. This socialization, coupled with inherent biological differences between the sexes (e.g., [81]), could cause the fear circuitry in females to be more susceptible to aberrancies induced by childhood maltreatment.

In the present study, females reporting high childhood maltreatment also exhibited evidence of greater fear generalization, as measured by fear-potentiated startle. These individuals demonstrated greater fear-potentiated startle responses to the CS− than did females reporting low childhood maltreatment. Additionally, females reporting high childhood maltreatment displayed statistically equivalent fear-potentiated startle responses to the CS+, GS1, GS2, GS3, GS5, GS6, and CS−, in contrast to all other groups. Collectively, these findings suggest that maltreated females may retain impaired long-term safety signal memory and a greater generalization of conditioned fear, specifically at the physiological level.

Greater fear generalization in maltreated participants is consistent with some prior work in the area. Lange et al. [12] and Thome et al. [44] both observed greater fear generalization in youth or adults who had experienced childhood maltreatment. In contrast to our findings, however, the evidence reported by these investigators was limited to subjective fear ratings and participant reaction time to generalization stimuli. Thus, our results are the first to show an association between childhood maltreatment and fear generalization as measured behaviorally via fear-potentiated startle. Moreover, previous work has not reported an association between childhood maltreatment and fear generalization that is sex-dependent. The sex-dependent nature of our findings could be explained by the findings of Herringa and colleagues [78]. In their study, maltreated females, but not males, exhibited lower hippocampus-sgACC and amygdala-sgACC functional connectivity. This “double hit” decoupling in females could cause them to experience less contextual and sensory gating of their fear expression, resulting in greater generalization.

There are some limitations of the present study that should be taken into consideration when interpreting the findings. First, our assessment of childhood maltreatment was retrospective. Thus, it is possible that participants did not precisely remember their childhood experiences, which could have altered the degree of childhood maltreatment that they reported. Second, although our categorization of “low” and “high” childhood maltreatment related to methodology employed in previous work, it is possible that this categorization misidentified some participants as experiencing “low” or “high” childhood maltreatment. Third, despite a large overall sample (*n* = 291), only 71 participants reported high childhood maltreatment. Combined with our primary effect being a higher-order interaction between childhood maltreatment and sex, the present findings should be considered preliminary in nature and in need of further replication.

## 5. Conclusions

The present findings suggest that childhood maltreatment is associated with sex-dependent effects on the acquisition and expression of conditioned fear. Specifically, females reporting high childhood maltreatment exhibited stronger fear learning and greater fear generalization than females or males reporting low childhood maltreatment. The sex-dependent nature of these findings is consistent with some existing research and should be examined more thoroughly in future studies. The neurobiological processes that underlie the observed alterations in females reporting high childhood maltreatment are not necessarily maladaptive. Indeed, in a child who endures chronic stress, it is adaptive for the brain to anticipate a high amount of stress throughout one’s lifespan and adjust its processing to enable rapid and efficient learning regarding new threats. Nevertheless, these seemingly adaptive processes can have maladaptive consequences and result in the expression of fear when no threat is present (see [82] for a related argument). Collectively, the findings presented here may prompt additional research that can inform clinicians about the mechanisms underlying female susceptibility to psychopathology.

## Figures and Tables

**Figure 1 brainsci-12-01536-f001:**
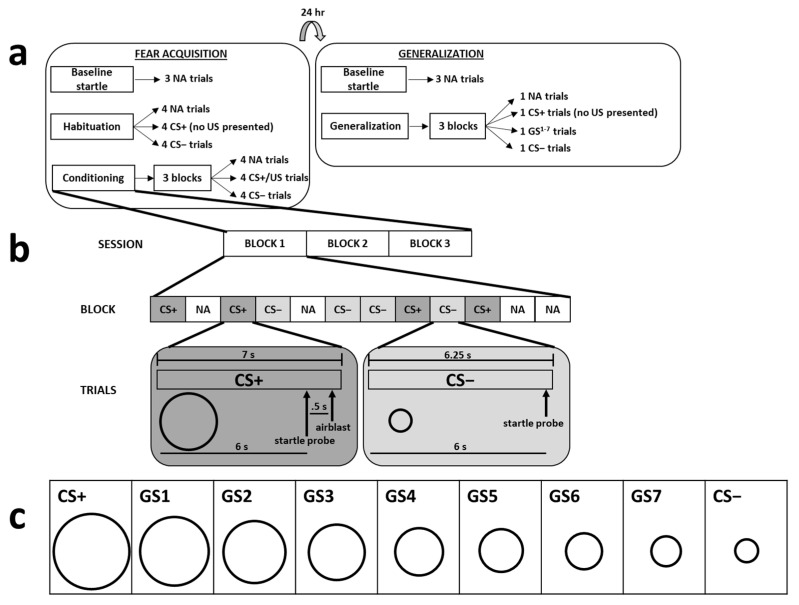
The top part of the figure depicts a summary of the 2-day fear-potentiated startle paradigm (**a**). The acquisition and generalization phases were separated by 24 h, and each phase began with 3 noise alone (NA) trials. Acquisition proceeded with a habituation phase, during which there were 4 NA, CS+, and CS− trials without any presentation of the US. Following habituation, participants underwent the conditioning phase, which consisted of 3 blocks of NA, CS+, and CS− trials, during which the CS+ was always paired with the US. The next day, generalization consisted of 3 blocks of NA, CS+, GS, and CS− trials; the US was not presented during any of these trials. The middle part of the figure depicts a diagram of the CS+ and CS− trials that occurred during the conditioning phase of acquisition (**b**). Participants were exposed to a total of 12 trials of 3 different trial types (NA, CS+, CS−) during each block. The timelines for stimulus exposure during the CS+ and CS− trials are depicted under these trial types. The lower part of the figure displays all 9 stimuli used during the generalization phase (**c**).

**Figure 2 brainsci-12-01536-f002:**
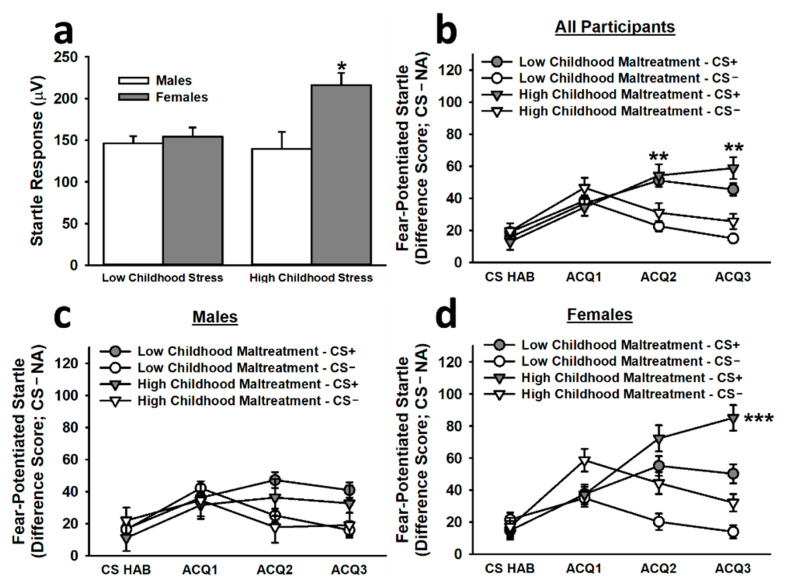
Baseline startle responses and fear-potentiated startle responses to the CS+ and CS− during the acquisition phase on Day 1. Females reporting high childhood maltreatment displayed significantly greater baseline startle responses than all other groups (**a**). The second inset (**b**) depicts the acquisition curve for differential fear conditioning; by the second and third blocks of reinforced acquisition trials, participants exhibited significantly greater fear-potentiated startle responses to the CS+ than to the CS−. Although both males (**c**) and females (**d**) exhibited significantly greater fear-potentiated startle responses to the CS+ than to the CS−, females reporting high childhood maltreatment demonstrated significantly greater fear-potentiated startle responses to the CS+ during late acquisition than all other groups. Data are means ± SEM. * *p* < 0.01 relative to all other groups; ** *p* < 0.001 relative to CS−. *** *p* < 0.01 relative to all other groups.

**Figure 3 brainsci-12-01536-f003:**
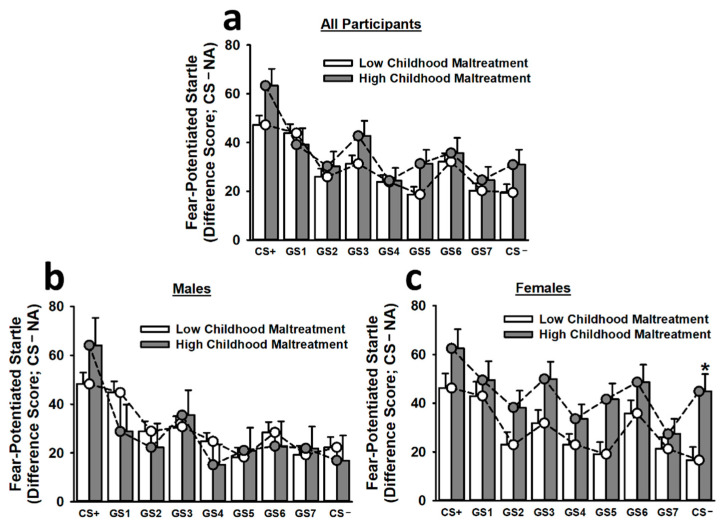
Fear-potentiated startle responses to the CS+, seven different GSs, and the CS− during the generalization phase on Day 2. The top figure inset (**a**) depicts fear-potentiated startle responses for all participants. Overall, participants exhibited a relatively smooth generalization gradient, with fear-potentiated startle responses being greatest to the CS+ and decreasing steadily as stimuli approached the CS−. No significant effect of childhood maltreatment on fear-potentiated startle responses was observed for males (**b**). Females reporting high childhood maltreatment demonstrated significantly greater fear-potentiated startle responses to the CS− than females reporting low childhood maltreatment (**c**). Moreover, females reporting high childhood maltreatment appeared to exhibit greater fear generalization, as, in contrast to all other groups, these participants displayed statistically equivalent fear-potentiated startle responses to the CS+, GS1, GS2, GS3, GS5, GS6, and CS−. Data are means ± SEM. * *p* < 0.001 relative to low childhood maltreatment.

**Figure 4 brainsci-12-01536-f004:**
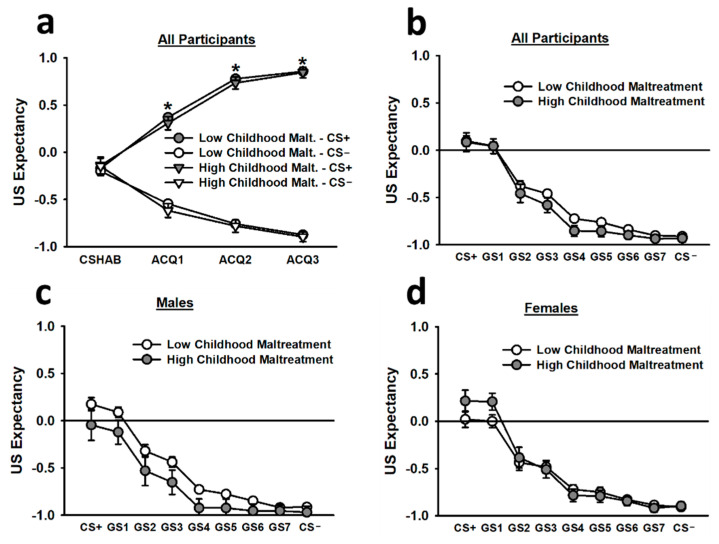
Subjective US expectancy ratings during the acquisition (**a**) and generalization (**b**–**d**) phases. According to the top left figure inset (**a**), participants demonstrated statistically equivalent US expectancy ratings for the CS+ and CS− during the non-reinforced CS habituation block (CS HAB). However, during the first, second, and third blocks of reinforced acquisition trials, participants exhibited significantly greater US expectancy ratings for the CS+ than for the CS−. During the generalization phase, participants, overall, displayed significantly greater US expectancy ratings for the CS+ and GS1 than for all other stimuli (**b**). This effect was observed in both males (**c**) and females (**d**). Data are means ± SEM. * *p* < 0.001 relative to CS−.

**Table 1 brainsci-12-01536-t001:** Self-report scores (means ± SEM) for low and high childhood maltreatment groups.

	Low Childhood Maltreatment (*n* = 220)	High Childhood Maltreatment (*n* = 71)
STAI—State Anxiety	38.28 (0.78)	42.81 (1.43) *
STAI—Trait Anxiety	39.07 (0.71)	43.30 (1.30) *
ASI	15.85 (0.76)	18.69 (1.39) ^β^
PCL-C	31.16 (0.76)	39.23 (1.41) **
CES-D	12.12 (0.62)	17.83 (1.14) **

* *p* < 0.01 relative to low childhood maltreatment; ** *p* < 0.001 relative to low childhood maltreatment; ^β^ *p* = 0.073 relative to low childhood maltreatment.

## Data Availability

The data reported in this manuscript are a subset of the data found in Collection 3016 of the NIMH Data Archive.

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
