# Peer review of "Childhood Maltreatment in Females Is Associated with Enhanced Fear Acquisition and an Overgeneralization of Fear"

_brainsci, 2022, doi:10.3390/brainsci12111536_

Round 1
Reviewer 1 Report
Dear Authors,
An interesting and captivating manuscript. The subject of fear is often untouched in the social sciences as it's considered taboo, but your work is part of the effort to reduce this. Bravo. Intricately put together, this manuscript is a very good example of academic conceptualization.
The strong points of your manuscript include:
1. A strong background/rationale for the research.
2. A detailed methodology that is well matched to the objectives of the research. Of particular note, the authors have done a very good job of citing the relevant sources to justify the uses of the tests/questionnaires for the research.
3. Strong discussion of the findings.
May I please make the following suggestions? Thank you.
1. It would benefit the reader to know exactly why the exclusion criteria (p. 3) were necessary to maintain the integrity of the data. May I please suggest that the authors explain how/why these exclusion criteria were necessary? Thank you.
2. The authors did not note any limitations faced in conducting this research. Perhaps the authors could share, for the benefit of the readers, the limitations they faced so as to better guide future research on this subject.
3. The authors have noted that females have higher fear acquisition than males when faced with childhood trauma. However, the authors have not attempted to explain why males are better able to manage this than females as per the findings of the study. Would the authors consider adding some postulations as to why this is so? Could it be cultural conditioning where males are acculturated/taught to be less fearful, or could it be that males due to cultural conditioning have been able to repress childhood maltreatment by explaining it away as part of socialization into being male? Thank you.
Overall, very good work and I wish the authors all the best in their future endeavors.
Author Response
Responses to Reviewer 1 Comments
We thank the reviewer for his/her comments. We believe that the changes made based on the comments have resulted in a substantially improved paper. Our point-by-point responses to the comments are below.
Point 1: It would benefit the reader to know exactly why the exclusion criteria (p. 3) were necessary to maintain the integrity of the data. May I please suggest that the authors explain how/why these exclusion criteria were necessary?
Response 1: We have included a brief explanation of the reasons for the exclusion criteria in the methods section.
Point 2: The authors did not note any limitations faced in conducting this research. Perhaps the authors could share, for the benefit of the readers, the limitations they faced so as to better guide future research on this subject.
Response 2: We have now included a section on limitations of the project in the discussion section.
Point 3: The authors have noted that females have higher fear acquisition than males when faced with childhood trauma. However, the authors have not attempted to explain why males are better able to manage this than females as per the findings of the study. Would the authors consider adding some postulations as to why this is so? Could it be cultural conditioning where males are acculturated/taught to be less fearful, or could it be that males due to cultural conditioning have been able to repress childhood maltreatment by explaining it away as part of socialization into being male?
Response 3: We thank the reviewer for raising this important point. Early socialization could indeed differentially impact male and female emotional expression and regulation. Most work suggests that females are socialized to be more emotionally expressive, which could result in greater overt fear-related behaviors. We have included an additional paragraph in the discussion to address this possibility.
Reviewer 2 Report
The paper was very thoroughly written in terms of its theoretical basis, inclusion criteria, description of the experiment and its results. A few points need clarification, especially regarding the classification of the subjects into two groups with low and high childhood stress. First, why the term trauma referring to abuse and neglect is further referred to stress. The introduction, on the other hand, refers to maltreatment. It is worthwhile to unify the terminology here. Secondly, the description of the division into groups according to the full 28-item scale is unclear. The authors cite a paper [58[ where there are three rather than five dimensions and each is described separately. The description of the method should be clear enough to apply such a division to other studies. Please provide the cut-off point or any other relevant rule.
The second comment concerns PTSD, which is given as an exclusion criterion, however it was analysed. PTSD and PCL-C are also used interchangeably.
It is not clear from the objectives of the study why the questionnaire part was included. This is the other thread that could be a separate paper. The place and method of surveying should be better described. It was a large battery of questionnaires, those burdened the respondents. There are too few references in the discussion to the results provided in Table 1 without references to other studies.
A drawback of the discussion is the lack of a section on the limitations. For example, one might mention a retrospective inquiry into childhood experiences, a small sample by gender.
The abstract also needs additions. It is worth adding the number of people classified di high stress group.
Although there is a section of the Institutional Review Board Statement in a footnote, a subsection "ethical considerations" would look good in the main body of the paper.
Editorial comments are: please clarify abbreviations in the footnotes under Table 1; on line 98, better use the word paper not manuscript; on line 94, please specify that this is the "young adults" group.
Author Response
Responses to Reviewer 2 Comments
We thank the reviewer for his/her comments. We believe that the changes made based on the comments have resulted in a substantially improved paper. Our point-by-point responses to the comments are below.
Point 1: First, why the term trauma referring to abuse and neglect is further referred to stress. The introduction, on the other hand, refers to maltreatment. It is worthwhile to unify the terminology here.
Response 1: We have replace all instances of “childhood stress” with “childhood maltreatment” for uniformity.
Point 2: Secondly, the description of the division into groups according to the full 28-item scale is unclear. The authors cite a paper [58] where there are three rather than five dimensions and each is described separately. The description of the method should be clear enough to apply such a division to other studies. Please provide the cut-off point or any other relevant rule.
Response 2: We thank the reviewer for pointing out this ambiguity. We have attempted to clarify the ambiguity of the classification by adding the cutoff values and underlining/italicizing the specific differences between the groups. Also, our classification is similar to previous work, but we included physical and emotional neglect in our classification, in addition to the abuse categories. We have added additional references that utilize a methodology similar to ours. The classification into high (moderate or severe levels of any abuse or neglect) and low (none or mild levels of all abuse and neglect) maltreatment has been shown to strongly predict depression and PTSD in past work.
Point 3: The second comment concerns PTSD, which is given as an exclusion criterion, however it was analysed. PTSD and PCL-C are also used interchangeably.
Response 3: We excluded individuals who had previously been diagnosed with PTSD to prevent adverse effects of the fear conditioning paradigm on their psychological health. However, we were still interested in assessing symptoms of PTSD in those individuals who participated. The PCL-C is a questionnaire that measures symptoms related to PTSD, but it is not used to diagnose the condition. Because this questionnaire is the one used for our assessment of PTSD symptoms, the term PCL-C is used to reflect that in the paper.
Point 4: It is not clear from the objectives of the study why the questionnaire part was included. This is the other thread that could be a separate paper. The place and method of surveying should be better described. It was a large battery of questionnaires, those burdened the respondents. There are too few references in the discussion to the results provided in Table 1 without references to other studies.
Response 4: The CTQ was included to enable an assessment of (and subsequent categorization based on) childhood maltreatment. The other questionnaires were used to verify that participants who reported high levels of childhood maltreatment also exhibited greater symptoms of current psychological distress. All of the questionnaires were 40 questions or less and did not take participants more than 10-15 minutes to complete. We only included references that point readers to the questionnaires themselves or to other work using the questionnaires. The primary focus of the paper is not on differences in symptoms of anxiety, PTSD, and depression in participants with low or high childhood maltreatment.
Point 5: A drawback of the discussion is the lack of a section on the limitations. For example, one might mention a retrospective inquiry into childhood experiences, a small sample by gender.
Response 5: We have now included a section on the limitations of the project in the discussion section.
Point 6: The abstract also needs additions. It is worth adding the number of people classified in the high stress group.
Response 6: This information has been added to the abstract.
Point 7: Although there is a section of the Institutional Review Board Statement in a footnote, a subsection "ethical considerations" would look good in the main body of the paper.
Response 7: We have added a section titled “ethical considerations” to the methods section.
Point 8: Editorial comments are: please clarify abbreviations in the footnotes under Table 1; on line 98, better use the word paper not manuscript; on line 94, please specify that this is the "young adults" group.
Response 8: The footnotes under Table 1 have been clarified. The word “manuscript” has been changed to “paper.” We have added the word “young” before “adults” in line 94.